# Rabies virus large protein-derived T-cell immunogen facilitates rapid viral clearance and enhances protection against lethal challenge in mice

Shimeng Bai [1,2,4], Xinghao Pan [3,4], Tianhan Yang[3,4], Nan Gao[3], Cuisong Zhu[3], Ai Xia[1], Meiqi Feng[1], Miaomiao Zhang[3], Xiaoyan Zhang [1,3] ✉ & Jianqing Xu [1,3] ✉

## Abstract

**Background** Rabies remains a devastating and fatal infectious disease worldwide. To date, vaccination is the most reliable and effective strategy for controlling rabies. However, despite the effectiveness of inactivated vaccines, cumbersome vaccination procedures and the high costs of post-exposure prophylaxis impose a significant economic burden, particularly in developing countries with limited access to vaccines. Therefore, there is an urgent need to develop a novel rabies vaccine that reduces costs while enhancing safety and efficacy. **Methods** We developed a novel mRNA rabies vaccine called RABV-G-LT, which incorporates two immunogens: RABV-G, a glycoprotein designed mainly to elicit neutralizing antibody responses, and RABV-LT, a T-cell immunogen derived from the large protein of the rabies virus. Additionally, we evaluated the immunogenicity of RABV-G-LT in both mice and non-human primates. **Results** The RABV-LT mRNA vaccination alone induced potent RABV-LT-specific T-cell responses and provided modest protection against rabies virus challenge in mice. Importantly, the dual-immunogen mRNA vaccine RABV-G-LT elicited vigorous and persistent neutralization antibody and T-cell responses, resulting in significantly more efficient clearance of the rabies virus in the brain and spinal cord. This conferred enhanced protection, evidenced by lesser initial weight loss and earlier recovery of body weight compared with the RABV-G mRNA or inactivated vaccine groups. Moreover, RABV-G-LT also mounted persistent strong antigen-specific T-cell and antibody immune responses in nonhuman primates. **Conclusions** Our study suggested that combining the T-cell immunogen and virus-neutralizing antibody immunogen was a practical approach to strengthening the defense against the rabies virus.

## Plain language summary

Rabies remains a devastating and fatal infectious disease globally. While current vaccines are effective, they require multiple administrations (shots) to achieve protection against the rabies virus. A more cost-effective vaccine could provide protection to more people. In this study, we develop a novel mRNA vaccine named RABV-G-LT and test its ability to activate the body's immune system and protect against the virus. We compare it to other strategies, including commercially available vaccines, and highlight that RABV-G-LT could strengthen the defense against the rabies virus and confer much-prolonged protection.

Rabies virus (RABV) is a neurotropic virus transmitted to humans primarily through the bite of an infected animal[1]. From the bite site, RABV travels along the peripheral nervous system to reach the central nervous system (CNS), and once clinical symptoms appear, the mortality rate approaches nearly 100%. However, RABV can remain at the entry site of infection for days or weeks before reaching the CNS. Hence, timely immunization, either before or after exposure, can effectively prevent the disease[2,3]. Thus, vaccination is required to trigger a prompt and effective immune response that

[1]Clinical Center of Biotherapy, Zhongshan Hospital & Institutes of Biomedical Sciences, Fudan University, Shanghai, P. R. China. [2]Bio-therapeutic Center, National Clinical Research Center for Infectious Disease, Shenzhen Third People's Hospital; The Second Hospital Affiliated with the School of Medicine, Southern University of Science and Technology, Shenzhen, China. [3]Shanghai Public Health Clinical Center & Institutes of Biomedical Sciences, Fudan University, Shanghai, P. R. China. [4]These authors contributed equally: Shimeng Bai, Xinghao Pan, Tianhan Yang. ✉e-mail: zhangxiaoyan@fudan.edu.cn; xujianqing@fudan.edu.cn

eliminates the rabies virus before it invades the CNS. To eradicate the rabies virus, we must focus on two key aspects: generating rabies virus-neutralizing antibodies (RV-nAbs) to neutralize the virus and triggering cellular immunity to clear virus-infected cells[4–7]. Current vaccinations primarily focus on inducing effective neutralizing antibodies, but cellular immunity is also crucial for controlling the spread of infection. Therefore, a more reliable and cost-effective vaccine that can induce early development of robust cellular immunity and RV-nAbs is needed.

With the rapid development of the mRNA vaccine field in the last decade, researchers have attempted to explore different mRNA rabies vaccine formulations to develop effective and cost-efficient vaccines. For example, the first unmodified mRNA rabies vaccine CV7201 evaluated in a phase I clinical trial[8], or CV7202 that changed the delivery material to lipid nanoparticles (LNPs)[9], or the recently reported novel mRNA rabies vaccine, LVRNA00[10], they were all designed to target the rabies virus glycoprotein (RABV-G), the surface protein of the virus, to induce effective RV-nAbs.

Rabies virus neutralization is antibody-dependent. However, cellular immunity is essential for controlling the spread of viral infection, especially in highly innervated areas of the body[11,12]. The cellular immune response is also strongly correlated with the duration and intensity of protection established by vaccination[13]. We hypothesized that a vaccine approach targeting a more conserved viral protein besides RABV-G would provide robust T-cell responses and confer enhanced protection. Among RABV viral proteins, the viral large protein L is a multifunctional protein responsible for virus transcription, replication[14,15], and post-transcriptional processing[16,17], and virus pathogenicity[18]. It is essential for the production of all viral proteins, including itself. RABV-L protein was also reported as the potential target for the treatment[18] or diagnosis of rabies[19]. In addition, bioinformatics analyzes have shown that RABV-L protein is more conserved across different strains compared with RABV-G (with amino acid sequence similarity exceeding 90%)[20]. Thus, RABV-L proteins as a novel T-cell immunogen could be incorporated into vaccine design to generate a more robust protective vaccine against rabies viruses.

In this study, we first developed a recombinant T-cell immunogen-based vaccine, named RABV-LT, derived from RABV large protein by using the bioinformatics approach to enrich the most conserved epitopes. We demonstrated that RABV-LT mRNA was highly immunogenic, able to elicit a robust RABV-LT-specific T-cell response, and modest control of rabies virus. Then, we combined mRNA RABV-LT with our previously developed RABV-G mRNA vaccine[21] to construct a dual-immunogen mRNA rabies vaccine RABV-G-LT. Then, we tested its protective efficacy against the lethal rabies virus infection. The data showed that the dual-immunogen mRNA vaccine induced a persistent cellular immune response and a humoral response in mice and nonhuman primates (NHPs). More importantly, the dual-immunogen RABV-G-LT mRNA rabies vaccine was more robust and effective in containing viral replication seven days after vaccination than RABV-G mRNA or commercial rabies vaccines. Our study provided an essential basis for the design of T-cell immunogens targeting the conserved regions of the RABV. Our data demonstrated that T-cell immunogens had significant potential to be introduced into the new rabies vaccine strategy.

## Methods
### Cells and virus
HEK293T cells (ATCC, CRL-3216) or BHK-21 cells (kindly provided by Dr. Zhang Shuye) were cultured with complete Dulbecco's modified Eagle's medium supplemented with 10% (*v/v*) fetal bovine serum and 1% penicillin–streptomycin at 37 °C in 5% $CO_2$ incubator. CVS-11 rabies challenge virus (GenBank: GQ918139.1) was generously provided by Wuxi Xin Lian Xin Biotech Co., Ltd (Wuxi, China).

### Analysis of functional T-cell epitopes of rabies large protein and identification of epitope-enriched fragments
Given the complexity of human leukocyte antigen (HLA) polymorphisms, 12 HLA-I and 54 HLA-II supertypes[22–24] were used in this study to predict restricted epitopes with maximum population coverage (covering more than 90% of selected HLA-I supertypes and 95% of selected HLA-II supertypes in the global population)[25], to trigger both CD4[+] and CD8[+] T cells for the development of potent and long-lasting immunity[22].

The NetMHC pan 4.1 and NetMHCII pan 4.0 servers were used to predict the location of potential HLA-I or HLA-II epitopes of rabies large proteins from five strains: PM1503 (accession number: DQ099525), CTN-1 (accession number: FJ959397), aG (accession number: GQ412744), ERA (accession number: EF206707), and CVS-11 (accession number: GQ918139). All potential epitopes with high-affinity binding (the thresholds were 2% ranked for HLA-I and 5% ranked for HLA-II) were selected for further analysis. Then, a custom computer-based algorithm was developed to analyze the enriched regions among the high-affinity binding epitopes. Meanwhile, the amino acid sequences from the five large rabies proteins were aligned through the Clustal Omega program[25]. The conservation score of amino acids in each position was also calculated. Next, the values of the corresponding region's strong-binding affinity epitope score multiplied by its amino acid conservation score were used as a criterion to intercept the T-cell epitopes of large proteins. Additionally, the original three to five amino acids were contained within the N and C-terminal peptide flanks to promote the efficient presentation of the intercepted T-cell epitopes[26]. Then, allergenicity and safety tests were conducted using AllerTOP V.2[27]、AlgPred 2.0[28]、AlgPred 2.0-IgE[29]、AllergenFP V1.0, and AllerCatPro 2.0[30]. Finally, the recombinant RABV-LT sequence was successfully constructed.

### Vaccines
The mRNA vaccines used in this study were nucleoside-modified mRNAs obtained by completely replacing the uridine triphosphate (UTP) with 1-methylpseudouridine-5'-triphosphate (Nanjing Synthgene Medical Technology Co., Ltd, China).

RABV-G mRNA vaccine expressing optimized rabies glycoprotein (PM, GenBank: AJ871962) was synthesized through in vitro transcription using a T7 high-yield RNA transcription kit following the manufacturer's protocols (Novoprotein, Shanghai, China) on linearized plasmid templates, as previously reported[21]. RABV-LT sequences were also cloned into the linearized plasmid template, similar to the method used for the RABV-G mRNA vaccine. Then, an internal ribosome entry site (IRES) from the encephalomyocarditis virus (EMCV) was introduced between the RABV-G and RABV-LT genes, creating a dual-immunogen mRNA, referred to as RABV-G-LT. The carboxy-terminal end of the RABV-G-LT gene includes an Escherichia coli dihydrofolate reductase (DHFR) domain, along with a FLAG tag for detecting RABV-LT expression. After transcription, the mRNAs were capped with the vaccinia capping system and an mRNA cap 2'-O-methyltransferase (Novoprotein, Shanghai, China), followed by purification using LiCl and 70% ethanol as we previously described[21]. The mRNAs were evaluated by agarose gel electrophoresis, and protein expression was detected in 293 T cells (see below). Finally, the mRNA-LNP vaccines were formulated by mixing the mRNAs in the aqueous solution and an ionizable cationic lipid mixture in ethanol using the previously described self-assembly method[31]. The concentration of the mRNA-LNP vaccines was calculated through a Quant-iT RiboGreen assay (Invitrogen, R11490).

The Inactivated Rabies vaccine in the study was obtained from Sigma–Aldrich (Product Code: EPR0100000, assigned the activity of 10 IU/vial, France).

### Protein expression and identification
RABV-G mRNA, RABV-LT mRNA, or RABV-G-LT mRNA was transfected into 293 T cells using Lipofectamine 3000 Reagent (Thermo Fisher Scientific, L3000001) to confirm in vitro expression. In brief, the cells were plated into six-well plates 12 h before transfection. Then, mRNAs and Lipofectamine 3000-mRNA were mixed in Opti-MEM (Gibco) in a ratio of 1:2 (4 µg mRNA vs. 8 µL of transfection reagent), and the mixture was added to each well. The cells transfected with RABV-G or RABV-LT mRNA were harvested 24 h later. For RABV-G-LT mRNA, the TMP132 (Sigma–Aldrich) was added to the cell culture medium 6 h before collection. All the cells were lysed with 4× sodium dodecyl sulfate loading buffer

(TaKaRa) for Western blot analysis. The primary antibody Rab-50 (Santa Cruz) was used to identify RABV-G (glycoprotein) protein, and the monoclonal ANTI-FLAG M2 antibody (Sigma–Aldrich) was used to detect RABV-LT protein, followed by peroxidase-conjugated goat anti-mouse immunoglobulin G (IgG) H&L (Yeasen, China).

## Animal experiments

Female BALB/c mice (6–8 weeks old, specific-pathogen-free) were purchased from Suzhou Hua Chang Biological Co., Ltd Female rhesus macaques (~3 years old) obtained from a domestic source (Ningbo, China) were all housed in the Shanghai Public Health Clinical Center (SPHCC) animal facility. The animal experiments were conducted according to the recommendations of the SPHCC Guide for the Care and Use of Laboratory Animals.

The immunization schedules were presented thoroughly in the relevant figures. For mouse immunization, all the mRNA vaccines were injected into the thigh of the hindlimb intramuscularly at a dose of 3 μg in a 100 μL volume. The mice in the negative control group were injected intramuscularly with empty LNP. Further, the mice were immunized using an inactivated vaccine intramuscularly (1 IU/mL). The sera or spleen samples were collected at different time points according to the immunization schemes shown for specific antibody analysis or T-cell responses. Following vaccination, the mice were challenged with 20 or 40 $MLD_{50}$ (mouse $LD_{50}$, 50 μL per mouse) of CVS-11 intramuscularly. Then, the body weight and survival were monitored daily. Three mice in each group were euthanized 7 or 12 days post-infection (dpi), and the spinal cord or brain tissues were collected to quantify viral loads and histopathological analysis (equivalent portions of the brain tissues).

Female rhesus macaques were randomly assigned to five groups ($N = 5$ in each vaccination group, $N = 4$ for empty-LNP negative control group): RABV-G mRNA (3 or 10 μg per macaque), RABV-G-LT mRNA (3 or 10 μg per macaque), and empty-LNP. All the vaccines (volume: 1 mL) were intramuscularly injected into the left deltoid muscle with a prime-boost regimen at 6-week intervals. Blood was collected at various times to isolate the serum and PBMCs (Peripheral Blood Mononuclear Cells).

## Antibody analysis

RABV-G-specific antibodies in mice or rhesus macaque sera were analyzed using ELISA. The RABV-G protein was obtained from AtaGenix (China) and precoated with 1 μg/mL. The sera were diluted 200-fold in the first well, followed by a twofold serial dilution. For IgM or total IgG, horseradish peroxidase (HRP)-conjugated Rabbit anti-Mouse IgM Ab (Abclonal, China) or HRP-conjugated anti-mouse IgG (Yeasen, China) at a 1:5000 dilution was added. For the subclass of antibodies, biotin-conjugated goat anti-mouse IgG1, IgG2a, and IgG2b (1:5000; Abcam) were incubated in the plates. Then, streptavidin-conjugated HRP (SA-HRP) (1:5000) (Yeasen) was incubated. Finally, the plates were colored using the substrate OPD (Sigma–Aldrich), and the absorbance at 490 nm was measured using a Synergy microplate reader (Bio-Tek). The endpoint titers were determined at the highest dilution, and the cutoff value was defined as yielding an OD value twice that of the empty LNP control. All the sera needed to be inactivated at 56 °C for 30 min to evaluate virus neutralization. The anti-RABV-neutralizing antibodies in the sera were determined using the FAVN (Fluorescent Antibody Virus Neutralization) test according to the World Health Organization protocol as described earlier[32].

## Enzyme-linked immunosorbent spot assay mice

The enzyme-linked immunosorbent spot (ELISPOT) assay was performed following the manufacturer's protocols (BD Bioscience). Briefly, the plates were precoated with anti-mouse interferon (IFN)-γ captured antibodies (5 μg/mL) overnight at 4 °C. Further, $2 \times 10^5$ splenocytes were added to the plates in duplicates and stimulated with the RABV-G peptide pools (5 μg/mL), RABV-LT peptide pools (5 μg/mL), or the media for 20–24 h at 37 °C. This was followed by incubation sequentially with biotin-conjugated mouse anti-IFN-γ antibody and alkaline phosphatase-conjugated SA.

Finally, the plates were colored using an AEC substrate reagent. The spots in plates were scanned and counted with a BioSpot plate reader (ChampSpot 437III; Beijing Sage Creation Science Co., Ltd).

## Macaques

ELISPOT assay was performed with the nonhuman primate IFN-γ kits (Mabtech) following the manufacturer's protocols. A total of $1 \times 10^5$ PBMCs were stimulated with RABV-G peptide pools (2 μg/mL), RABV-LT peptide pools (2 μg/mL), or the media as negative control. The plates were incubated at 37 °C for 48 h. The spots were scanned and analyzed using an ELISPOT reader, as mentioned earlier.

Peptide-specific spots were determined after the deduction of spots of the negative control wells (wells without peptides) from spots of RABV-G peptides or RNBA-LT peptides. Results were considered as the average number of peptide-specific spots in replicate wells.

## Flow cytometry assay

Antigen-specific T-cell response was analyzed using intracellular cytokine staining (ICS). Briefly, $1 \times 10^6$ isolated splenocytes of mice were suspended in phosphate-buffered saline containing 0.2% bovine serum albumin and counted, followed by stimulation with RABV-G peptide pools (1 μg/mL), RABV-LT peptide pools (1 μg/mL), or the media. Golgi-Plug (BD Biosciences) was added at 1:1000 dilution after 1 h. The cells were stained with Amcyan LIVE/DEAD Aqua (BioLegend) and incubated with antibodies against CD3 (Percpcy5.5, Clone 17A2; BioLegend), CD4 (AF700, Clone RM4-5; BD Biosciences), and CD8 (fluorescein isothiocyanate, Clone 53-6.7; BioLegend). After fixation and permeabilization (BD Biosciences), the intracellular staining was employed through staining with antibodies against anti-IFN-γ (PE, Clone XMG1.2; BD Biosciences), anti-IL-2 (APC, Clone JES6-5H4; BioLegend), or anti-TNF-α (BV605, Clone MP6-XT22; BioLegend). The cells were collected using BD FACSAria III flow cytometer (BD Biosciences) and then analyzed using FlowJo 10 software.

## Viral RNA extraction and RT-PCR quantification

The brain or spinal cord tissues were collected using the RNAzol RT Reagent (Molecular Research Center) following the manufacturer's protocols. Then, the total RNA was extracted using the RNA isolation kit (Direct-zol RNA Miniprep Plus kit; Zymo Research). The concentration was measured using the NanoDrop reader (BioTek). One-step reverse transcription–polymerase chain reaction (RT-PCR) was carried out using the HiScript II One-Step qRT-PCR SYBR Green Kit (Vazyme) and a Bioer real-time PCR system to compare rabies viral RNA copies in each group.

The oligo primers used for rabies nucleoprotein were as follows:

Forward: 5′-AATGCGACGGTTATTGCTGC-3′; reverse: 5′-TGCCACGTCGGTCTTTGTTA-3′.

The RT-PCR cycling was performed as follows: 50 °C for 15 min, 95 °C for 2 min, followed by 40 cycles of 95 °C for 10 s and 60 °C for 30 s. The relative fold change was calculated using the $2^{-\Delta\Delta Ct}$ method and normalized based on the non-infection control (blank group).

## Histopathological assay and RNAscope in situ hybridization (ISH) assay

The brain tissues from mice were collected, fixed immediately with 4% ($v/v$) paraformaldehyde, and embedded in paraffin to sagittally cut paraffin sections with a 4–5 μm thickness. Following deparaffinization, the histopathological changes in the brains of mice in the infected group were examined using hematoxylin and eosin staining and viewed under a light microscope.

RNAscope in situ hybridization (ISH) was performed to analyze rabies viral RNA expression in brain sections. The viral nucleoprotein (NP) RNA served as the target due to its high conservation across different rabies virus strains. For this assay, the NP-specific RNAscope probe (V-RABV-gp4 (220268) from ACDBio) was employed to detect viral RNA. The RNA ISH assay was conducted using the RNAscope Multiplex Fluorescent Reagent Kit V2 (Advanced Cell Diagnostics, 323100).

## Statistical analysis

In the study, the data were presented as mean ± SEM. The titer Data were shown as GMT ± geometric SD. One-way ANOVA followed by Tukey's multiple comparisons tests and Kruskal–Wallis (survival data) was used for statistical analysis using GraphPad Prism (Version 9.4). $P$ values < 0.05 ($^*P < 0.05$, $^{**}P < 0.01$, $^{***}P < 0.001$, $^{****}P < 0.0001$; ns, not significant) indicated statistically significant differences.

## Ethics statement

All animal experiments in this study were approved by the Institutional Animal Care and Use Committee (IACUC) of Shanghai Public Health Clinical Center. Macaque studies and related experimental procedures were approved by the Laboratory Animal Welfare and Ethics Committee of Shanghai Public Health Clinical Center (SPHCC) (approval number: 2022-A038-01).

All mouse experiments were conducted after approval by the Laboratory Animal Welfare and Ethics Committee of Shanghai Public Health Clinical Center (SPHCC) (approval number: 2021-A073-02). All infection experiments were performed in the ABSL2 laboratory following guidelines of Environmental Health and Safety. All work was performed with approved standard operating procedures and safety conditions for the RABV. The procedures used for anesthesia and euthanasia of study animals followed tenets of the ARRIVE reporting guidelines. During the RABV challenge study, the mice with body weight drop more than 30%, severe paralysis, or inability to feed were euthanized using carbon dioxide inhalation.

## Reporting summary

Further information on research design is available in the Nature Portfolio Reporting Summary linked to this article.

## Results

### Construction and immunogenicity of RABV large protein-derived T-cell immunogen

T-cell epitopes from RABV large protein (L) of five virus strains were predicted by bioinformatics analysis. We identified potential epitopes presented by 12 common HLA-I alleles for human CD8$^+$ T cells (MHC-I) and 54 HLA-II alleles for human CD4$^+$ T cells (MHC-II) (prevalent in the human population[26], Supplementary Information, Table S1) using NetMHC pan 4.1 and NetMHCII pan 4.0 servers. The binding threshold parameters were 2% and 5% for MHC-I and MHC-II, respectively. We then analyzed the spatial distribution of high-affinity peptides (Fig. 1A and Supplementary Data 5). As illustrated in Fig. 1A, the distribution of high-affinity epitopes in the RABV large protein of PM1503 strain was depicted under various HLA phenotypes (in the figure, the red lines indicate the locations of strongly binding epitopes). Then, we identified the enriched regions through a comprehensive analysis of the distribution of high-affinity binding sites in large proteins among the five virus strains mentioned earlier and the conservation score of amino acids in these regions (Supplementary Data 6), and assembled as a RABV Large protein-derived T-cell immunogen, named RABV-LT protein. In addition, we analyzed the homology of RABV-LT protein epitope sequences using the SimPlot + +[33] and found that they were highly conserved across the virus strains, as mentioned earlier, with amino acid similarity of almost more than 90% (Fig. 1B). This was consistent with a previous study[20].

We first generated a methyl-pseudouridine-modified (m1Ψ) mRNA that encoded the codon-optimized RABV-LT protein to evaluate the immunogenicity of recombinant RABV-LT antigen (Fig. 1C, top). The synthesis, purification, and LNP formation of the mRNA vaccine were conducted as previously described[21]. Western blot analysis confirmed the proper expression of the RABV-LT protein in 293 T cells after RABV-LT mRNA transfection (Fig. 1C, bottom, Supplementary Information, Fig. S1).

The immune responses induced by the RABV-LT mRNA vaccine were analyzed in BALB/c mice. Two groups of mice ($N = 5$ per group) were vaccinated intramuscularly with RABV-LT mRNA (3 µg) or empty-LNP

(without mRNA) as a negative control on day 0 (Fig. 1D). The mice were euthanized and subjected to immunological analyzes 1 week after prime vaccination. First, T-cell responses were examined in splenocytes by IFN-gamma intracellular staining (ICS) based flow cytometry assay. The RABV-LT-specific T-cell responses were determined after stimulation of splenocytes with a 15-amino-acid peptide pool that spanned the part of RABV-LT protein (L660 aa–Q950 aa) (Supplementary Information, Table S2). The RABV-LT mRNA vaccine induced high RABV-LT-specific CD4$^+$ and CD8$^+$ T-cell responses compared with those in the empty-LNP group [$P < 0.05$ for two cytokines, except for interleukin (IL)-2; Fig. 1E, F]. RABV-LT-specific T cells appeared to predominantly express interferon-γ (IFN-γ, mean, 0.60% for CD4$^+$ T cells and 1.25% for CD8$^+$ T cells), followed by tumor necrosis factor-α [(TNF-α); Fig. 1E, F]. The RABV-LT-mRNA vaccine–induced T-cell responses were also evaluated by an IFN-γ ELISPOT assay (Fig. 1G). Compared with the negative control, the vaccine elicited significant IFN-γ$^+$ spots in the splenocytes of the RABV-LT-vaccinated group [Mean spot-forming cells (SFCs)/10$^6$ splenocytes, 825 versus 58; $P < 0.01$; Fig. 1G].

To answer whether vaccination with the RABV-LT vaccine alone would provide effective protection from rabies virus infection, we evaluated the protection efficacy of RABV-LT mRNA in animal challenge models. BALB/c mice ($N = 10$ per group) received either one RABV-LT mRNA vaccination or control were intramuscularly challenged with 20 MLD$_{50}$ of CVS-11 in a volume of 50 µL at day 7 post-immunization. Body weights and survival were monitored over the following 2 weeks. All mice ($N = 10$) died in the control empty-LNP immunized group within 9 days, whereas 70% of the mice survived in the RABV-LT mRNA immunized group (Fig. 1I, $P < 0.0001$). After 4 dpi, mice in the control group experienced a rapid decrease in body weight, whereas only less than 10% weight loss was observed in the RABV-LT mRNA vaccinated group (Fig. 1H). These findings suggested that modest protection could be achieved by T-cell immune responses alone induced by the RABV-LT mRNA vaccine against the lethal rabies virus challenge.

### Dual immunogen vaccine of RABV-G and RABV-LT elicited robust, specific T-cell and humoral immunity

After the RABV-LT mRNA vaccine was confirmed immunogenic, we further tested whether combining RABV-G with RABV-LT into a dual-immunogen vaccine would produce more potent immune protection in mice. Therefore, we produced an M1-modified mRNA vaccine that expressed both RABV-G and LT protein. The IRES from EMCV was introduced between the RABV-G and RABV-LT genes to ensure the co-expression of both genes from a single mRNA transcript. Then, *E. coli* dihydrofolate reductase (DHFR) was genetically fused to the carboxyl-terminal of the RABV-LT molecule (Fig. 2A). As is known, DHFR is an inherently destabilizing domain degraded by proteasomes unless it is stabilized by its high-affinity ligand trimethoprim (TMP). HEK293T cells were transiently transfected with RABV-G-LT mRNA to verify the expression of the new construct RABV-G-LT, with/without TMP treatment at 50 µM TMP for 6 h before the harvest of cells. The expression of RABV-LT protein increased in the presence of TMP compared with that in the absence of TMP (Fig. 2B and Supplementary Information, Fig. S2).

We next determined the immunogenicity of the dual-immunogen RABV-G-LT mRNA vaccine in mice (Fig. 2C). Two groups of mice ($N = 5$ per group) were either vaccinated with the dual-immunogen RABV-G mRNA vaccine. (3 µg) or the empty LNP as control at week 0. Then, ELISPOT and ICS assays were performed 10 days or 20 weeks post-immunization. The mouse splenocytes isolated at various time points were restimulated with RABV-LT or RABV-G peptide pools (Supplementary Information Table S3). The data showed that dual-immunogen RABV-G-LT mRNA vaccination induced robust RABV-G-specific (Fig. 2E, F) and RABV-LT-specific (Fig. 2G, H) CD4$^+$ and CD8$^+$ T-cell responses, respectively. Obviously, TNF-α was highly expressed in both RABV-G- and RABV-LT-specific T cells, followed by IFN-γ and IL-2 (Fig. 2E–H). IFN-γ ELISPOT analyzes also confirmed the significant induction of IFN-γ in the

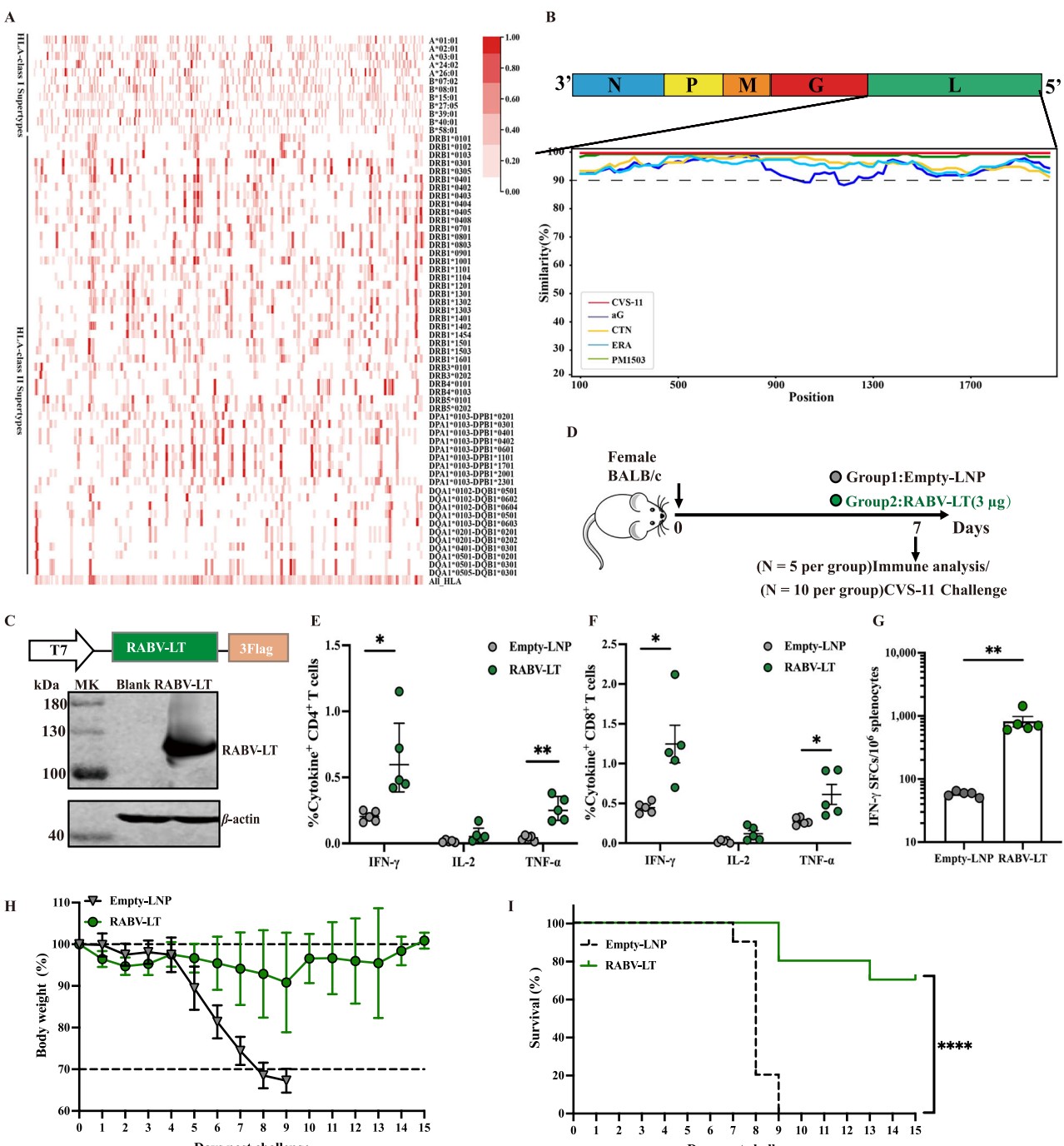

**Fig. 1 | Construction and immunogenicity of recombinant RABV-LT.**
**A, B** Bioinformatics prediction of potential T-cell epitopes for RABV large protein. **A** Distribution of RABV-derived HLA-I and HLA-II supertype representative epitopes with the high binding affinity derived from the large protein sequence. Red bars show strong–binding affinity epitopes for each HLA supertype molecule with 2% and 5% rank to HLA-I and HLA-II supertypes, respectively. **B** Similarity plot based on the full-length genome sequence of RABV large protein. Genome sequences of PM1503 (accession number: DQ099525), CTN-1 (accession number: FJ959397), aG (accession number: GQ412744), ERA (accession number: EF206707), and CVS-11 (accession number: GQ918139) were compared with CVS-11 (GQ918139). **C** Schematic illustration of recombinant RABV-LT design schemes (top) and Western blot of the recombinant RABV-LT mRNA expressed by HEK293T cells (bottom). **D** Experimental design and schedule. A total of 30 mice were included in this study. Two experiments were conducted; one was to measure vaccine-induced T-cell responses on day 7 ($N = 5$ per group), and another was to assess the protective efficacy of the RABV-LT vaccine against rabies virus infection ($N = 10$ per group). **E, F** ICS measurements of RABV-LT-specific CD4[+] and CD8[+] T cells in the mouse spleen (day 7) are shown. The percentages of individual cytokine-positive CD4[+] (**E**) or CD8[+] (**F**) T cells were compared between the empty-LNP (negative control) or mRNA RABV-LT mRNA vaccine groups. **G** IFN-γ ELISPOT measurements of antigen-specific T cells in the spleen (day 7) are shown. Mice ($N = 10$) were challenged with 20 MLD$_{50}$ standard virus CVS-11 intramuscularly. Body weight relative to day 0 (**H**) and survival percentage (**I**) were recorded daily during a 14-day observation period. The data are presented as SFCs per $10^6$ splenocytes and as mean ± SEM. One-way ANOVA, followed by Tukey's multiple comparison tests (**E–G**) and the Kruskal–Wallis test (**I**), was used for statistical analysis. $^*P < 0.05$, $^{**}P < 0.01$, $^{****}P < 0.0001$.

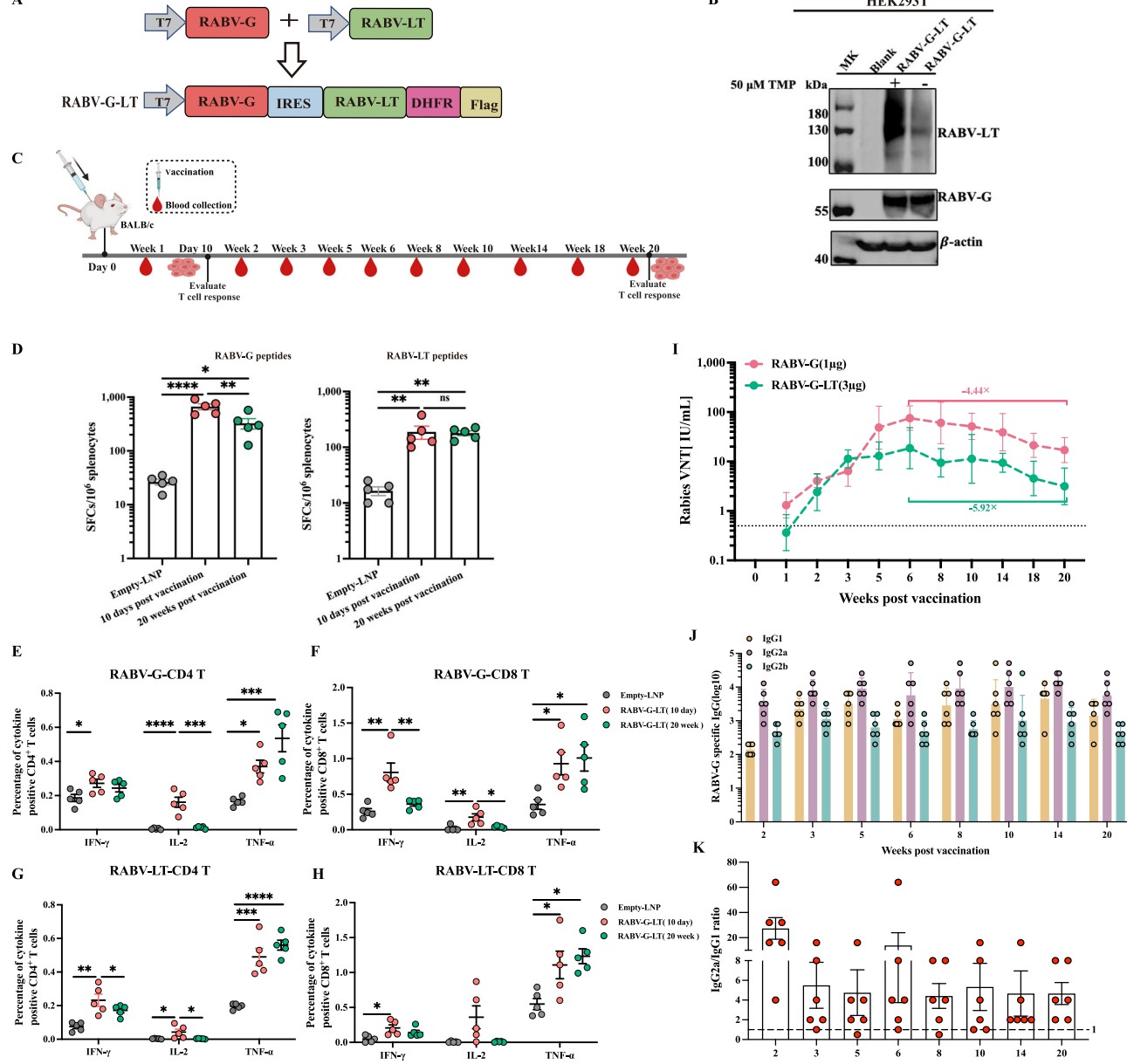

**Fig. 2 | Combining RABV-G with RABV-LT vaccination elicited specific T-cell and humoral immunity.** Dual-immunogen RABV-G-LT vaccine–induced T-cell responses and long-term humoral responses were measured following vaccination. **A** Schematic illustration of dual-immunogen RABV-G-LT design schemes. **B** Western blot of the recombinant RABV-G-LT mRNA expressed by HEK293T cells. **C** Mice immunization design and timeline. A total of 30 mice were included in this study. **D** IFN-γ ELISPOT measurements of RABV-G-specific (left) or RABV-LT-specific (right) T cells in the spleen (10 days or 20 weeks) are shown ($N = 5$). The data are shown as IFN-γ SFC per $10^6$ splenocytes. **E, F** ICS analysis of RABV-G-specific CD4[+] and CD8[+] T cells in the mouse spleen. **G, H** ICS analysis of RABV-LT-specific RABV-G-specific T cells at different time points (10 days versus 20 weeks post-immunization, mean = 665 or 327; $P < 0.01$) (Fig. 2D, left). Further-more, the RABV-LT-specific IFN-γ-secreting T-cell responses remained consistent for at least 20 weeks (Fig. 2D, right).

CD4[+] and CD8[+] T cells in the mouse spleen. **I** Serum was collected at different time points for 20 weeks and analyzed for the VNA titers in either the RABV-G mRNA (1 μg, $N = 5$) or dual-immunogen RABV-G-LT mRNA (3 μg, $N = 6$) vaccinated group. The levels of IgG1, IgG2a, and IgG2b RABV-G-specific antibodies in sera (**J**) and the ratios of IgG2a/IgG1 (**K**) were assessed using ELISA in the RABV-G-LT mRNA-vaccinated group. The data are presented as mean ± SEM. The titer data were shown as GMT ± geometric SD. One-way ANOVA, followed by Tukey's multiple comparisons tests (**D–H**), was used for statistical analysis. *$P < 0.05$, **$P < 0.01$, ***$P < 0.001$, ****$P < 0.0001$.

A sustained, effective antibody response is one of the main goals of rabies vaccination. To analyze the longitudinal humoral immune responses, two groups of BALB/c mice were intramuscularly inoculated with either RABV-G (1 μg, $N = 5$) or the dual-immunogen RABV-G-LT mRNA (3 μg,

$N = 6$) at week 0, and the blood samples were collected at different time-points (Fig. 2C). Due to RABV-G-LT being three-fold in molecular weight than RABV-G, thus we used 3 μg mRNA for the RABV-G-LT vaccine while only 1 μg mRNA was administrated for the RABV-G vaccine in order to generate similar G proteins in vivo. The levels of RABV-virus-neutralizing antibodies (RV-nAbs) were measured. As early as in the first week following vaccination, the RABV-specific neutralizing titers reached or exceeded 0.5 IU/mL in all mice in the RABV-G mRNA group, while 4 mice out of 6 in

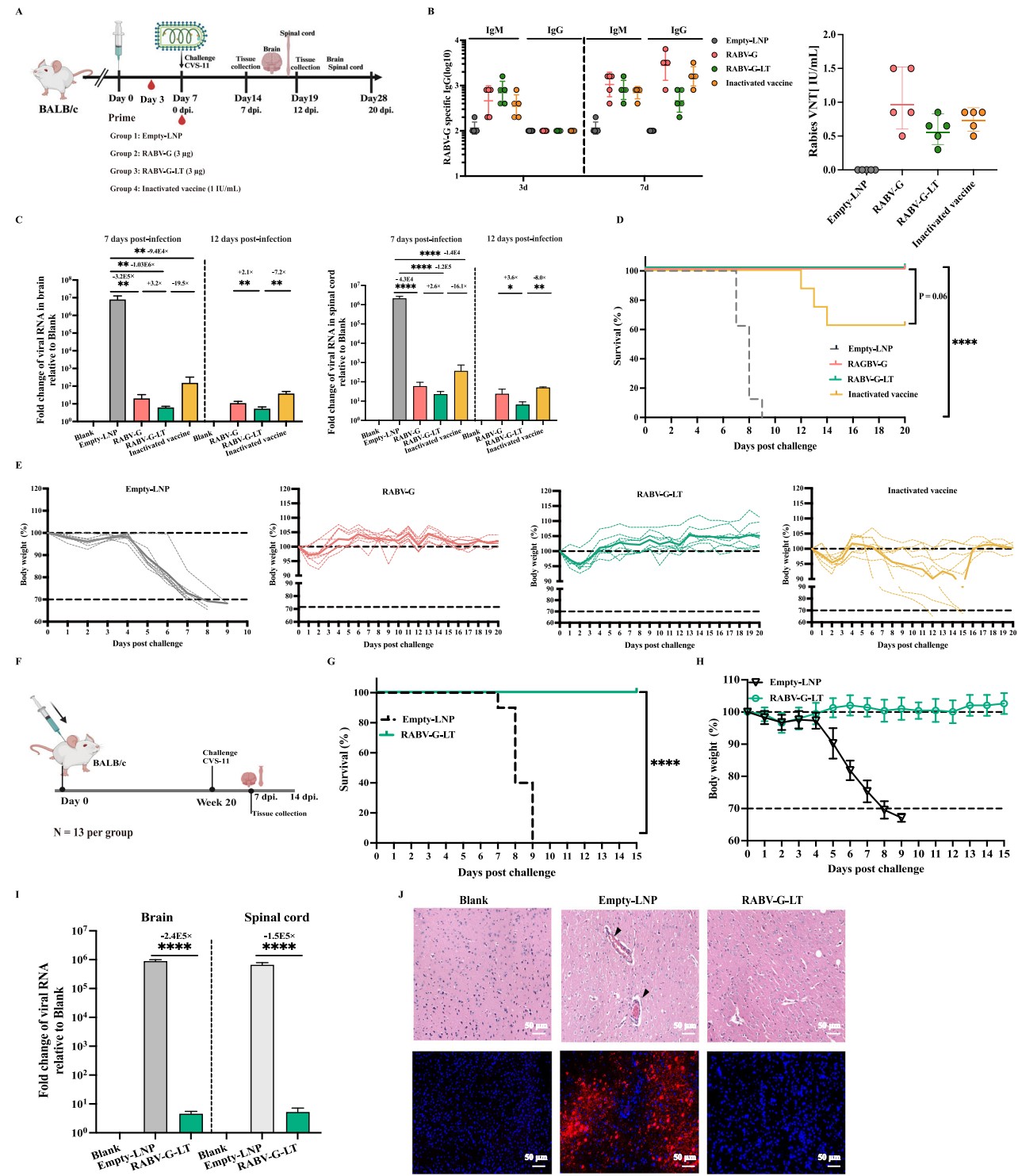

the RABV-G-LT mRNA group did so (Fig. 2I). Then, VNA titers increased to 11.3 IU/mL at week 3, peaked at week 6 (18.7 IU/mL), then dropped to 3.2 IU/mL at week 20 in the RABV-G-LT mRNA group, with a decrease of 5.92-fold from the peak VNTs (Fig. 2I). Interestingly, VNA titers in the RABV-G mRNA group decreased by 4.44-fold from their peak VNTs (75.4 IU/mL). Then, we also measured the IgG isotypes using ELISA. The RABV-G-LT immunized mice mounted high RABV-G-specific IgG2a (beneficial for Th1-biased immune response) levels at 2–20 weeks following vaccination (Fig. 2J and K).

Our data suggested that RABV-G-LT vaccination triggered both vigorous T-cell responses and humoral responses, promoting a predominantly

Th1-biased immune response. These features may strengthen the defense against the rabies virus.

### RABV-G-LT vaccination promoted viral clearance and immune protection against rabies virus infection in mice

After demonstrating that RABV-G-LT mRNA vaccination could elicit antigen-specific immune responses, we next investigated whether the dual-immunogen vaccine would provide strong protection against lethal RABV challenge during early immunization in BALB/c mice. Four groups of mice (N = 14 per group) were immunized with empty-LNP (negative control), RABV-G mRNA (3 μg), dual-immunogen RABV-G-LT mRNA vaccine

**Fig. 3 | Combining RABV-G with RABV-LT vaccination conferred improved protection against rabies virus infection compared with RABV-G vaccination alone in mice.** A total of 56 mice were included in this study. Four groups of BALB/c mice ($N = 14$) were intramuscularly vaccinated with empty-LNP (as a negative control), mRNA RABV-G vaccine (3 µg), mRNA RABV-G-IT vaccine (3 µg), and inactivated vaccine (1 IU/mL; as a positive control) on day 0. Then, on day 7, all mice were challenged with CVS-11 (**A**–**E**). **A** Experimental design and schedule. **B** Immune responses assessed within the first 7 days post-vaccination are shown ($N = 5$). **C** Each group of three mice was sacrificed at 7 or 12 dpi, at which point the brains and spinal cords were removed to evaluate the expression of nucleocapsid viral RNA (measured using RT-qPCR and normalized based on the non-infection control). **D, E** Changes in the body weight and survival curves of mice in each group following challenge in 20 days. Survival after CVS-11 challenge (**D**). The mean weight in each vaccinated cohort is indicated with a thick colored line; the weights of individual mice are indicated with a colored dashed line (**E**). **F** Mice immunization design and timeline for long-term monitoring. **F**–**J** Mice ($N = 13$) were challenged with $40 MLD_{50}$ (50 µL per mouse) CVS-11 intramuscularly 20 weeks following vaccination. Body weight (**G**) and survival (**H**) were recorded daily for 2 weeks. **I** Then, on day 7 following infection, the brains and spinal cords (three mice in each group) were collected to detect the expression of nucleocapsid viral RNA (measured using RT-qPCR and normalized based on the non-infection control). **J** The sagittal sections of mouse brains were stained for histopathological analysis and RNAscope in situ hybridization (ISH) assay of Brain tissues from 7 days following infection; representative histological changes are shown (scale bar = 50 µm). The black triangles in the figure indicate typical pathological changes, including vascular inflammatory cuffing (perivascular cuffing) and/or intravascular coagulation. Representative images of in situ hybridization (ISH) illustrated viral nucleoprotein (NP) expression in brain tissue. Each red dot corresponds to a single NP RNA molecule, with nuclei counterstained by DAPI. The data are presented as mean ± SEM. One-way ANOVA was used for statistical analysis, followed by Tukey's multiple comparison tests (**B, C, I**) and the Kruskal–Wallis (**D, H**) test. $^*P < 0.05$, $^{**}P < 0.01$, $^{****}P < 0.0001$.

(3 µg), or inactivated vaccine (1 IU/mL; as a positive control) on day 0, followed by intramuscular challenge with the CVS-11 strain ($40 MLD_{50}$) on day 7. Three mice per group were euthanized on day 7 or 12 post-infection, respectively, for viral load analyzes in the brain or spinal cord. The left eight mice were monitored for their survival up to 20 days after the challenges (Fig. 3A).

Additionally, we assessed the immune responses during the first 7 days post-vaccination (Fig. 3A). We observed that RABV-G-specific IgG antibody titers were low in all groups on day 3 (Fig. 3B, left). By day 7, all mice vaccinated with the RABV-G mRNA vaccine (3 µg) exhibited RABV-specific neutralizing antibody titers that reached or exceeded 0.5 IU/mL, with a geometric mean titer (GMT) of 0.96 IU/mL. This response was higher than that of the inactivated vaccine (positive control), which had a GMT of 0.7 IU/mL (Fig. 3B, right). For the dual-immunogen RABV-G-LT mRNA vaccine (3 µg), only one mouse showed a neutralizing titer below 0.5 IU/mL at day 7 (Fig. 3B, right). Then, the mice were challenged with lethal RABV on day 7. As expected, the empty-LNP administered group exhibited abundant RABV RNA on 7 dpi (As all mice in the empty-LNP group died before 12 dpi, the quantifications of viral RNA in this group were implemented only on 7 dpi), whereas all the vaccination groups had extremely low but detectable levels of RABV RNAs in both the brain and spinal cord (Fig. 3C). Interestingly, our results demonstrate that both mRNA vaccine constructs outperformed the inactivated vaccine in promoting the reduction of viral RNA levels in the brain and spinal cord at different time-points. Specifically, the RABV-G-LT mRNA vaccine group exhibited an impressive ~20-fold reduction in viral RNA levels in the brain at 7 days post-inoculation (dpi), and a 7.1-fold reduction at 12 dpi, when compared to the inactivated vaccine group (Fig. 3C, left). Similarly, the RABV-G mRNA vaccine also showed a viral RNA reduction in the brain or spinal cord; however, it was less effective, with reductions of only 3.2-fold and 2.1-fold at 7 and 12 dpi, respectively, in comparison to the RABV-G-LT mRNA vaccine (Fig. 3C, left). A similar trend was observed in the spinal cord, further reinforcing the superiority of the mRNA vaccine constructs over the inactivated vaccine (Fig. 3C, right). Accordingly, animals in the empty-LNP group underwent a rapid weight loss after 4 dpi (Fig. 3E) and experienced death during days 7–9 after the CVS-11 challenged (Fig. 3D). In contrast, both mRNA vaccine-immunized animals showed complete survival (100%), whereas five of eight (~62.5%) animals in the inactivated vaccine immunization group survived (Fig. 3D). Furthermore, RABV-G-LT mRNA group subjected only a transient minor weight loss and then experienced a weight gain during the subsequent 20-day observation, although RABV-G mRNA group experienced a similar transient weight loss when compared with RABV-G-LT mRNA group during the early infection, this group almost failed to gain any weight during the 20-day follow-up (Fig. 3E). Interestingly, the animals in the inactivated vaccine group showed maximal weight loss, and about 40% animals could not recover after challenge (Fig. 3E). We conducted a 20-day observation in this experiment across all three vaccination groups. The surviving mice usually returned to their original body weights within 14 days, then

continued to further increase from 14 to 20 days post-viral challenges in the absence of any clinical signs of rabies virus infection during this period. However, an extended observation up to clearance of viruses will be more convincing to reach a solid conclusion for protective efficacies.

To determine whether the RABV-G-LT mRNA vaccine can confer prolonged protection, the BALB/c mice ($N = 13$ per group) either received one immunization of RABV-G-LT mRNA or empty-LNP control and were intramuscularly challenged with $40 MLD_{50}$ of CVS-11 in a volume of 50 µL at 20 weeks. The survival and body weights were monitored daily for the following 14 days (Fig. 3F). All mice died in the empty-LNP group, whereas the mice vaccinated with RABV-G-LT mRNA once survived, demonstrating the potent efficacy of RABV-G-LT mRNA against RABV infection (Fig. 3G, H, $P < 0.0001$). In addition, the analyzes of viral RNA levels in the brain or spinal cord (three mice per group) on 7 dpi also demonstrated that RABV-G-LT mRNA vaccination could reduce viral RNA to extremely low levels (Fig. 3I). Consistent with the viral load analyzes, the brain histopathological analysis showed that CVS-11 infection caused evident changes in the empty-LNP group, including intravascular coagulation and perivascular cuffing. In contrast, no lesions were observed in the RABV-G-LT mRNA vaccination group (Fig. 3J top). Correspondingly, we identified abundant dot signals of RABV-specific viral RNAs in the brains of empty-LNP mice, but no viral RNA in brain sections from vaccinated animals (Fig. 3J bottom).

Overall, these findings demonstrated that the dual-immunogen RABV-G-LT vaccine could not only generate strong and quick protection from RABV infection in the brain and spinal cord during early vaccination, but it could also achieve prolonged protection for at least 20 weeks following vaccination. Importantly, the dual-immunogen RABV-G-LT mRNA vaccine could launch a more efficient viral clearance and stronger protection than the single immunogen RABV-G mRNA vaccine, even though it mounted less neutralization antibody responses, heightening the importance of mounting T-cell responses in addition to humoral responses alone.

**Dual-immunogen RABV-G-LT vaccination elicited persistent RABV-G- and RABV-LT-specific immunity in nonhuman primates**

Next, we evaluated the immunogenicity of RABV-G-LT in nonhuman primates, rhesus monkeys. Four groups of monkeys ($N = 5$ per group) were vaccinated intragluteally with either RABV-G mRNA or dual-immunogen RABV-G-LT mRNA vaccine at a dose of 3 or 10 µg, respectively. As a negative control, another group of monkeys ($N = 4$) was injected with empty-LNP (Fig. 4A). A single immunization with RABV-G-LT mRNA at a dose of 3 or 10 µg produced the GMT VNTs less than 0.5 IU/mL, whereas the immunization with RABV-G mRNA induced the VNTs more than 0.5 IU/mL in all monkeys within 4 weeks following prime vaccination (Fig. 4B, C). Then, we tested the potency of the vaccines to boost immune responses. A second vaccination with the same dose as the prime vaccination was performed at week 6. The levels of VNTs increased at least nearly 100-fold in the RABV-G-LT mRNA group and 10-fold in the RABV-G

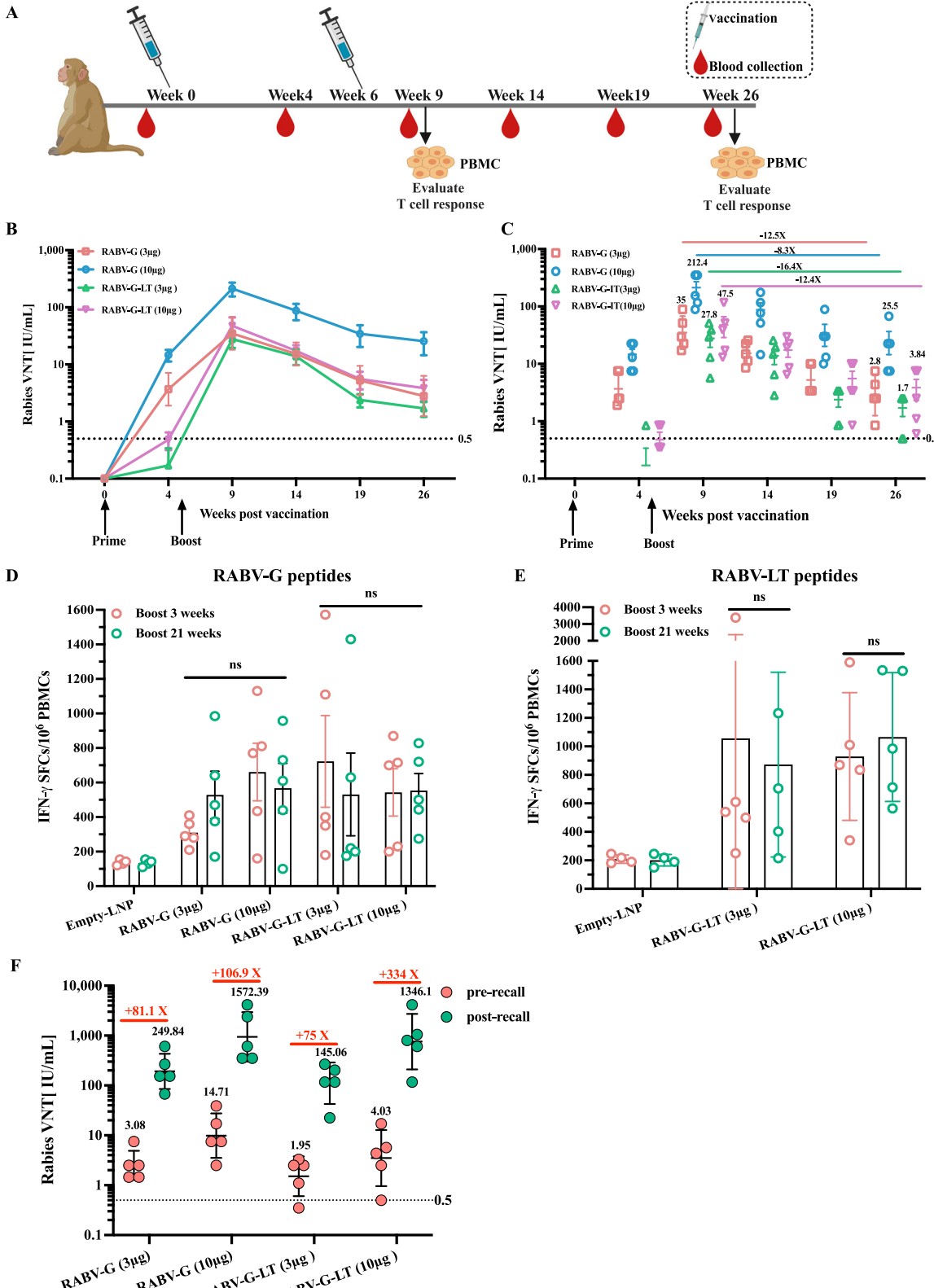

**Fig. 4 | Dual-immunogen RABV-G-LT vaccination elicited persistent RABV-G-and RABV-LT-specific immunity in nonhuman primates.** RABV-G-LT vaccination induced strong humoral and cellular responses in NHPs. Rhesus monkeys ($N = 5$) were randomly categorized into four groups and intragluteally injected with either RABV-G mRNA or dual-immunogen RABV-G-LT mRNA vaccine at a dose of 3 or 10 µg, respectively. The monkeys in the negative control group ($N = 4$) were injected with empty-LNP. **A** Rhesus monkey immunization design and timeline. **B**, **C** Kinetics of virus-neutralizing antibody titers in the plasma. Monkey IFN-γ

ELISPOT measurements of RABV-G-specific (**D**) or RABV-LT-specific (**E**) T cells in PBMCs (week 9 or week 26). PBMCs were stimulated with either RABV-LT or RABV-G peptide pools. The data are shown as IFN-γ SFCs per $10^6$ PBMCs. **F** VNA titers in plasma were measured before and 2 weeks after the recall vaccination, and administered 7 months after the second vaccination. One-way ANOVA, followed by Tukey's multiple comparison tests (**D**, **E**), was used for statistical analysis. Ns not significant.

mRNA group 3 weeks after the second vaccination (Fig. 4B, C). The kinetics of induced humoral response following the second vaccination in 20 weeks were monitored. Following an initial decline, the neutralizing antibody titers stabilized at above 0.5 IU/mL for the two doses (in the RABV-G group, 3 µg vs 10 µg, GMT = 2.8 or 25.5; in the RABV-G-LT group, 3 µg vs 10 µg, GMT = 1.7 or 3.84) in either RABV-G mRNA or RABV-G-LT mRNA group (Fig. 4B, C). Additionally, we measured the T-cell responses induced by the RABV-G-LT or the RABV-G mRNA vaccines in NHPs. IFN-γ ELISPOT analysis of PBMCs was performed in week 9 or 26. Unlike the humoral responses, which had a 12- to 16-fold decline in 20 weeks, the RABV-specific-G or -LT T-cell responses were not subject to a significant decline in 9 or 26 weeks in the RABV-G-LT mRNA group, which was also consistent with the antigen-specific T-cell responses in mice (Fig. 4D, E). This phenomenon was similar in the RABV-G mRNA group (Fig. 4D, E). Finally, we administered a third vaccination (recall vaccination) to the same animals 7 months later. Two weeks after the booster vaccination, we measured neutralizing antibody titers. In both vaccine groups, especially the RABV-G-LT group, we observed a rapid 74-fold increase in VNTs at a dose of 3 µg compared with 330-fold at a dose of 10 µg, demonstrating that the RABV-G-LT mRNA vaccine induced a robust recall response (Fig. 4F). We also found following the third vaccination, the neutralizing antibody titers showed no difference between the RABV-G and RABV-G-LT group.

## Discussion

Rabies remains a devastating infectious disease globally, causing significant morbidity and mortality, especially in developing countries with limited vaccine access[1]. The currently available inactivated rabies vaccines, such as the 5-dose Essen regimen (1-1-1-1-1) and the 4-dose Zagreb regimen (2-1-1), effectively induce immunity. However, to achieve protective immunity, multiple-dose regimens are required[10,21,34–38], thus imposing a substantial economic burden and limiting their use in low-resource settings[39,40]. Therefore, developing a novel rabies vaccine that reduces costs while enhancing safety and efficacy is critical.

Recent advancements in mRNA vaccine technology offer a promising approach for developing a more effective rabies vaccine. This study introduces a novel non-replicating mRNA vaccine that encodes both the rabies virus glycoprotein (RABV-G) and a recombinant RABV-LT protein derived from the large protein (L). We first confirmed that RABV-LT is a strong immunogen, inducing T-cell responses and providing modest protection against lethal rabies virus. Then, by combining the RABV-LT T-cell immunogen with RABV-G mRNA, we developed the dual-immunogen vaccine RABV-G-LT. This vaccine showed superior control of lethal rabies virus infection in the brain and spinal cord compared to the inactivated vaccine or RABV-G mRNA alone, particularly in the early stages of vaccination. Additionally, RABV-G-LT facilitated significant recovery in body weight among infected animals, highlighting its therapeutic potential.

By targeting highly conserved epitopes within the L protein, we aimed to induce robust T-cell immune responses that enhance viral clearance. Utilizing bioinformatics, we identified conserved epitopes from the L protein across various RABV strains and created a chimeric immunogen incorporating these elements. Our results confirmed that RABV-LT mRNA was immunogenic and elicited strong antigen-specific CD8+ T-cell responses, providing protection against lethal challenges. Our vaccine design emphasizes the importance of T-cell responses in combating viral infections, as these can prevent viral replication and eliminate infected cells. This concept is also supported by studies on other viral pathogens, including influenza[41] and SARS-CoV-2[42,43].

We further explored whether combining RABV-LT with RABV-G could provide improved protection against RABV by incorporating an internal ribosome entry site (IRES) to promote the co-expression of RABV-G and RABV-LT from a single transcript. Additionally, a destabilizing domain sequence known as dihydrofolate reductase was located downstream of the RABV-LT sequence. This sequence has been shown to facilitate the rapid degradation of RABV-LT within the proteasome, enhancing epitope generation and antigen presentation to dendritic cells[44], thereby generating effective T-cell responses against infectious viruses[45,46]. Our investigation revealed that the dual-immunogen RABV-G-LT provided stronger viral control and faster recovery from weight loss compared to inactivated vaccine or RABV-G mRNA, indicating enhanced protective efficacy. Importantly, the vaccine induced a Th1-biased immune response, critical for clearing virus-infected cells, reinforcing the notion that T-cell immunity is vital in defending against rabies.

Current vaccines are administered post-exposure to prevent disease onset, necessitating a prompt and robust immune response. Traditional inactivated vaccines have been enhanced with adjuvants, such as Toll-like receptor agonizts to accelerate immune responses[7,12], or Mn J rabies vaccines to enhance both humoral and cellular immunity[47]. Our findings suggest that the bivalent RABV-G-LT mRNA vaccine could enable rapid production of virus-neutralizing antibodies (VNAs) and strong T-cell responses as early as seven days post-vaccination, making it a potentially effective strategy for rabies prevention.

Prolonged protection is essential for any vaccine. Our data demonstrated that the RABV-G-LT mRNA vaccine fully protected mice against lethal RABV challenges 20 weeks after a single vaccination, with RABV-LT-specific T-cell responses persisting beyond this period. Interestingly, while the molar dosage of RABV-G was the same, a 3 µg dose of the dual-immunogen RABV-G-LT vaccine resulted in diminished RABV-G-specific antibodies compared to a 1 µg dose of RABV-G alone. This suggests that RABV-LT responses may compete with RABV-G, reducing its specific antibodies. Understanding the regulatory mechanisms of RABV-G and LT-specific immunity following vaccination is crucial.

To evaluate the vaccine's performance in larger animals and humans, we tested the RABV-G-LT mRNA vaccine in non-human primates (NHPs). NHPs vaccinated twice at a 6-week interval with 3 or 10 µg doses of RABV-G or RABV-G-LT mRNA exhibited persistent antibody responses (titers above WHO-recommended thresholds) for at least 7 months. Notably, a third dose significantly enhanced VNA titers and reduced variability between the two vaccines, consistent with previous findings[48]. Assessing the RABV-G-LT mRNA vaccine's effectiveness in NHPs will help translate results more comprehensively to humans. Additionally, evaluating its efficacy in other animal models, such as hamsters, ICR mice, beagles, and pigs, will provide further insights.

In conclusion, our study presents a promising new approach to rabies vaccination by combining the strengths of both humoral and cellular immunity through a novel mRNA vaccine platform, highlighting the urgent need for accessible and effective rabies prevention strategies globally.

## Data availability

All data of this study are available in the paper and/ or the Supplementary Materials. The source data for each figure is in the following Supplementary Data files: Fig. 1, Supplementary Data 1; Fig. 2, Supplementary Data 2; Fig. 3, Supplementary Data 3; Fig. 4, Supplementary Data 4; The distribution of high-affinity epitopes on the large protein among rabies virus (MHC I, MHC II result), Supplementary Data 5; Amino acid conservation rate in the L proteins among five virus strains, Supplementary Data 6; Other data are in the Supplementary Information file, including Supplementary Figs. S1–S3, Supplemental Tables S1–S3.

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

## Acknowledgements

This work was supported by the National Natural Science Foundation of China (No. 82071788), Innovation Action Plan Supported by Shanghai Science and Technology Commission (21S11903600), and the technology Service Platform for Detecting High-level Biological Safety Pathogenic Microorganism Supported by the Shanghai Science and Technology Commission (21DZ2291300).

## Author contributions

B.S. performed most of the experiments and analyzed data. P.X. designed the T-cell immunogen sequences, Y.T. conducted the mice sample collection and parts of mice or monkey experiments, X.A., Z.C., G.N., and Z.M. helped perform the ELISA or Virus-neutralization measurement. F.M. helped conduct the T-cell analysis. B.S. and P.X. wrote the manuscript. X.J. and Z.X. conceived and guided the study, and X.J. revised the manuscript. All authors reviewed and approved the manuscript.

## Competing interests

All the authors declare no competing interests. Dr. Jianqing Xu, Dr. Xiaoyan Zhang, and Ms. Shimeng Bai are applying for a patent from the Chinese Patent Office, which is relevant to this study.
