## [Transparent Peer Review file · Communications Medicine]

Rabies virus Large Protein-derived T-cell immunogen facilitates rapid viral clearance and enhances protection against lethal challenge in mice

Corresponding Author: Professor Jianqing Xu

Version 0:

Reviewer comments:

Reviewer #1

(Remarks to the Author)

Dear authors,

Thank you for this interesting manuscript. I believe that the approach of the development of a vaccine working both on the humoral and cellular immune responses against rabies is very valuable and would be an added value to the existing vaccines.

I do have however some remarks on the manuscript:

- 1) on line 285 you mention that 10 mice were used per group, however in figure 1 it is stated that 15 mice were used. How many animals were used?
- 2) Which route of administration was used for the RABV-LT mRNA vaccination, intramuscularly? Was it injected in the same paw or the opposite one? If the same paw was used, would it be possible that the protection is due to a local reaction rather than systemic? Especially since the challenge was performed so soon after vaccination.
- 3) compared to the other challenge studies a lower dose of CVS-11 was used in the first experiment, was there a specific reason for this?
- 4) on Line 334 it states "RABV-G-LT immunized mice showed significantly higher RABV-G-specific IgG2a levels" : As compared to which group? In none of the mentioned graphs are other groups included that we e.g. vaccinated with commercially available vaccines.
- 5) You mention that mice are followed up 20 days post inoculation, this seems short especially since virus is present in the brain and spinal cord of these animals. Have you performed test to for viability of the virus in these tissues? Can you be sure that none of the animals would have developed lethal infection later than 20 DPI.
- 6) line 348-350: a total of 14 mice was included per group, 3 were euthanized for viral load and 8 were followed up for survival, but 8+3 does not equal 14...
- 7) small typo: line 351-352 "as all mice died BEFORE 12 dpi"

Reviewer #2

(Remarks to the Author)

Shimeng Bai and co-authors present the results of their studies on the development of an mRNA rabies vaccine. Here they report on the development of a T-cell immunogen designed from the L protein which they also combine with a previously designed mRNA targeting the RABV G protein. Both constructs protected mice from lethal challenge. Both in mice and NHP the G protein mRNA produced more virus neutralizing antibodies than the G-LT construct. The co-authors argue, based on the evidence presented, that despite this the G-LT construct is a better vaccine. I have a few major concerns that must be addressed.

Major comments:

1. I could not find the LT sequence that was used in the construct reported. To allow replication experiments, the sequence must be reported.
2. (Line 51) In arguing that a better rabies vaccine is needed the co-authors have neglected one of the largest hurdles to accessing proper PEP which is rabies immune globulin. Current vaccines require passive immunity to protect and clear the

virus while the adaptive response is stimulated. If a vaccine could be designed that does not require passive immunity, then this would remove the current problems of cost and availability of rabies immune globulin. As such examining the immune response within the first 7 days (e.g. day 2-4 post vaccination) would be needed to support this argument.

3. In the same way many of the comparisons were between the mRNA constructs when the comparison should be to current vaccines, if the co-authors wish to argue that their vaccine is a needed improvement.

4. (Line 150) While the groups sizes varied the total number of mice used should be stated. Additional information regarding euthanasia criteria and euthanasia methods must be included for animal welfare. Were the animal experiments reviewed by an independent committee similar to IACUC?

5. The discussion must be rewritten. It is currently too long and repetitive. The co-authors need to focus the discussion on their arguments and the supporting evidence.

Minor comments:

6. (Line 41) This statement is confusing. I believe the co-authors are referring to the onset of rabies symptoms once the virus reaches the CNS. In fact, the virus does not replicate rapidly after infection which is correctly stated in line 43.

7. (Line 50) Rabies PEP is highly effective and current vaccines are focused on inducing virus neutralizing antibodies because VNA is effective in preventing disease.

8. (Line 114) Were allergy and safety testing conducted in animals?

9. (Line 127) No mention of the FLAG tag.

10. (Line 131) add (see below) for protein expression and detection

11. (Line 174) What expression system was used to make the G protein? Is it native, denatured, monomer, trimer?

12. (Line 236) staining for RABV antigen would be more specific and informative than H&E

13. (Line 356) Any animal euthanized with RABV in its brain it is difficult to say whether the animal is recovering or virus is just beginning to replicate and the animal will succumb. I think that including the current inactivated rabies vaccine for this comparison would be more informative since some animals did succumb in this group. Also this is a more important comparison than just comparing the two mRNA constructs.

14. (Line 419-20) Please cite evidence of poor immunogenicity of current rabies vaccines. The multi-dose regimen is more a product of the history of rabies PEP starting with Pasteur's 21-dose regimen than any scientific evidence.

15. (Line 480 and 499) I would suggest the hamster rabies PEP model as a well-established model for further testing. Dogs and pigs should only be used if the vaccine is intended for vaccination of these species.

16. Figure 1B the colors don't match the legend so it is difficult to distinguish which line is which.

17. Figure 1D add Immune analysis n = 5 / CSV-11 Challenge n =10 (Line 640) correct n=15

Version 1:

Reviewer comments:

Reviewer #1

(Remarks to the Author)

Thank you for the administered corrections, I do however still have some minor remarks/comments

Line 49-50: This line was corrected, however the way it is stated right now implies that the current vaccines are unreliable, which is not the case shown by the number of people surviving thanks to PEP treatment. It would be an advantage to develop a vaccine which can develop sufficiently high antibody titers after single shot, and this would be an improvement, but I do not believe that you can state the current vaccines are unreliable.

Line 57-58: The rabies virus "infects" the central nervous system, it does not paralyze it. Paralysis is a potential result of the infection.

Line 67-70: Taken into account that it takes approximately 7-10 days for the immune system to elicit an immune response upon vaccination, do you really believe that a vaccine could be developed that makes the administration of immunoglobulines unnecessary? Passive immunoglobulines are administered at the side of exposure immediately or as soon as possible after exposure. With vaccines it would still take 7-10 days to have antibodies that can react...

Line 463: Although I can understand that mRNA vaccines might elicit a higher immune response, I do not agree with stating that the current vaccines are poorly immunogenetic. They also elicit high and longlasting immune responses. I believe that the focus should be more on the fact that multiple-dose regimens results in therapy failure because of availability

Line 500-502: when being exposed to a potentially rabid animal, post-exposure, including HRIG should be administered as soon as possible. Waiting 7 days for an immune response via vaccination leaves the virus uncontrolled at the site of exposure. Do you really believe that this would be a safe solution?

Reviewer #2

(Remarks to the Author)

Version 2:

Reviewer comments:

Reviewer #1

(Remarks to the Author)

Dear Authors,

Thank you for your answers. I have not additional remarks.

Kind regards,

Point-by-point Response to Reviewers' Comments

Reviewer #1: Thank you for this interesting manuscript. I believe that the approach of the development of a vaccine working both on the humoral and cellular immune responses against rabies is very valuable and would be an added value to the existing vaccines.

Responses: We thank the reviewer for his/her encouraging comments.

1. on line 285 you mention that 10 mice were used per group, however in figure 1 it is stated that 15 mice were used. How many animals were used?

Response: We apologize for this confusion. In our mice studies, we immunized 15 mice per group, 5 out of 15 mice were sacrificed first to assess T-cell responses, the left 10 mice per group were then used to determine the protective efficacy of RABV-LT mRNA vaccine against rabies viral challenge. Therefore, the discrepancy arises because Figure 1D reflects the total of 15 mice used in the mice study.

To ensure the figure precisely reflects the details provided in the text. We revised Figure 1D as below.

The description and figure legends have also been revised accordingly.

2. Which route of administration was used for the RABV-LT mRNA vaccination, intramuscularly? Was it injected in the same paw or the opposite one? If the same paw was used, would it be possible that the protection is due to a local reaction rather than systemic? Especially since the challenge was performed so soon after vaccination.

Responses: We thank the reviewer for this insightful comments and questions regarding our study.

The RABV-LT mRNA vaccine was administered intramuscularly in one hind leg on

day 0, and the rabies virus challenge was performed in the opposite hind leg on day 7. Meanwhile, we also assessed T-cell responses in splenocytes on day 7 post-immunization. Our results demonstrated that RABV-LT mRNA vaccine effectively induced high levels of RABV-LT-specific T-cell responses, supporting that the observed protection is attributable to a systemic immune response rather than a localized reaction.

3. compared to the other challenge studies a lower dose of CVS-11 was used in the first experiment, was there a specific reason for this?

Responses: The lower dose of CVS-11 used in the first experiment was chosen to establish a baseline for assessing vaccine efficacy under less severe challenge conditions. This approach allowed us to evaluate the vaccine's performance and determine its protective threshold before using higher doses in subsequent experiments. The results informed our dose selection strategy for further studies, ensuring robust evaluation of vaccine efficacy across different challenge levels.

4. on Line 334, it states, "RABV-G-LT immunized mice showed significantly higher RABV-G-specific IgG2a levels": As compared to which group? In non of the mentioned graphs are other groups included that we e.g. vaccinated with commercially available vaccines.

Responses: The statement " RABV-G-LT immunized mice showed significantly higher RABV-G-specific IgG2a levels " was indeed an oversight, as we did not include comparisons with other vaccine groups in this experiment. Accordingly, we have carefully reviewed the manuscript and revised the manuscript to correct this mistake.

“The RABV-G-LT immunized mice mounted high RABV-G-specific IgG2a (beneficial for Th1-biased immune response) levels at 2–20 weeks following vaccination (Fig. 2J and 2K).” (**Lines 356-358**, revised manuscript).

5. You mention that mice are followed up 20 days post inoculation, this seems short especially since virus is present in the brain and spinal cord of these animals. Have you performed test to for viability of the virus in these tissues? Can you be sure that non of the animals would have developped lethal infection later than 20 DPI.

Responses: We observed that the mice returned to their original body weights within 14 days, then gradually increased from 14 to 20 days, without showing any clinical signs of rabies virus infection during this period. Therefore, we consider it is unlikely that there would be any additional deaths after 20 days. However, we acknowledge that an extended observation up to clearance of viruses will be more convincing. We appreciate the reviewer to point out this limitation and we added one additional sentence to remind readers as following: “We conducted a 20-day observation in this experiment across all three vaccination groups. The survived mice usually returned to their original body weights within 14 days, then continued to further increase from 14 to 20 days post viral challenges in the absence of any clinical signs of rabies virus infection during this period. However, an extended observation up to clearance of viruses will be more convincing to reach a solid conclusion for protective efficacies.”(Lines 403-408, revised manuscript)

6. line 348-350: a total of 14 mice was included per group, 3 were euthanized for viral load and 8 were followed up for survival, but 8+3 does not equal 14.

Responses: We apologize for this confusion. 14 mice were initially included per group, then 3 mice on day 7 and 3 mice on day 12 post-infection were euthanized for viral load analyses, and the left 8 mice were followed up for survival assessment. We have revised the manuscript to reflect the number of mice used and their allocation across the experimental endpoints.

“Three mice per group were euthanized on day 7 or 12 post-infection, respectively, for viral load analyses in the brain or spinal cord. The left eight mice were monitored for their survival up to 20 days after the challenges (Fig. 3A).” (Lines 370-372, revised manuscript).

7. small typo: line 351-352 "as all mice died BEFORE 12 dpi"

Responses: We thank the reviewer’s suggestion. Accordingly, we have revised accordingly:

“As expected, the empty-LNP administrated group exhibited abundant RABV RNA on 7 dpi (As all mice in the empty-LNP group died before 12 dpi, the quantifications of viral RNA in this group were implemented only on 7 dpi).” (Lines 380-383,

revised manuscript).

Reviewer #2: Shimeng Bai and co-authors present the results of their studies on the development of an mRNA rabies vaccine. Here they report on the development of a T-cell immunogen designed from the L protein which they also combine with a previously designed mRNA targeting the RABV G protein. Both constructs protected mice from lethal challenge. Both in mice and NHP the G protein mRNA produced more virus neutralizing antibodies than the G-LT construct. The co-authors argue, based on the evidence presented, that despite this the G-LT construct is a better vaccine.

Response: We thank the reviewer for his summary of our main findings.

1. *I could not find the LT sequence that was used in the construct reported. To allow replication experiments, the sequence must be reported.*

Response: We apologize for not including the LT sequences used in our construct. To address this, we provide the LT sequences in the supplementary material, and it will be shown in the revised manuscript.

The LT sequences are shown as follows:

“5’-MTDNCSRSYKVLKDYFKKVDGGTAAQSMVSLWLCGAHRSRRCITDLA
HFYSKSSPKAFGRYLANTYSSYLFFHVITLYMNALDQIWGLLIVTKDFVYSQS
SNCLFDRNYTLMKDLFLSRFNLSMILLSPEPRYSDDLISQLCQFPMFIKDKV
NQLEGTFPSAKRFFRVLDQFDNIHDLVFVYGCYRHDLARRILRWGFDKYSWY
LDSRFLARDHPFEIPESMDSHSFTRTRLKERELKIEGRFFALMSWNLRLYFVIT
EKLLANYILPLFDALMTKLIDRVTGQLLDYSRVTYAFHLDYEKWESTEDV
FSVLDQVFGLKRVFSRTHEFFQKSWIYYSDRSDLIGLRQKGWSLLYELESISR
NALSIRAIKKEETMCSYDFLIYGKTPLFRGNILVPMSTVSTNAKPMRDFLLM
SVQAVFHLLFSPILKGRVYKILSAEGFLLAMSRIIYLDPSLLLKTHRDNFILF
LKSVELFPRFLSELFSSSFLGIPQNSRTIRRQFRKSLSRRTLEESGNPRVSVSVLPS
FDQSFGYLGSSTSMSTQLFHAWEKTNVHVVKRALSLKESINAPVFKRTGSAL
HRFKSAQNGKNYDFMFQPLMLYAQTWTSELVQRDTRHIGSAQGLLYSILVAI
HDSGYGTIFPVNIYILIGSSICFRPLELISGVISYILLRLDNHPSLYIMLREPSLRG
EIFSIPQKREGNRSILCYLQHVLRYEREAFSDFRSVKMTYLTLMTYQSHLLLQ

RVERMRATLRQMSSLMRQVLGGHALSKRFQNPGLRVRVAVLNMFPDSKL
VFNSLLVNDLMASGTHPLPPSAIRYFQSVQKQVNMSYDLICDEVTDIASINRI
TLLMSDFALSIDGPLYLVFKTYGTMLVNPDYKAIRAFPSVTGFVTQVTSSFSSE
LYLRFKRGKFFRDAEYLTSSTLREMSLVLFNCSSKSEMQRARSLNYQDLVR
LSKVAIIISIMIVFSNRVFNISKPNICCSTMMYLSTALGDVPSSSLSLSSHWIRLI
YKIVKTSGEVERHLHGYNRWITLED-3'.”

2. (Line 51) *In arguing that a better rabies vaccine is needed the co-authors have neglected one of the largest hurdles to accessing proper PEP which is rabies immune globulin. Current vaccines require passive immunity to protect and clear the virus while the adaptive response is stimulated. If a vaccine could be designed that does not require passive immunity, then this would remove the current problems of cost and availability of rabies immune globulin. As such examining the immune response within the first 7 days (e.g. day 2-4 post vaccination) would be needed to support this argument.*

Response: We thank the reviewer for this instructive suggestion. To address this point, we added several sentences to describe this urgent needs (**Lines 67-72** in revised manuscript) and also examined the immune responses within the first 7 days post-vaccination.

“In addition, the largest hurdle is to access proper PEP which is rabies immune globulin. Current vaccines require passive immunity to protect and clear the virus while the adaptive response is stimulated. If a rabies vaccine could be designed that is able to elicit early immune responses and thereby does not require passive immunity, then this would remove the current problems of cost and availability of rabies immune globulin.” (**Lines 67-72**, revised manuscript)

Additionally, our data, detailed below, indicated the followings: “Additionally, we assessed the immune responses during the first 7 days post-vaccination (Fig. 3A). We observed that RABV-G-specific IgG antibody titers were low in all groups on day 3 (Fig. 3B, left). By day 7, all mice vaccinated with the RABV-G mRNA vaccine (3 µg) exhibited RABV-specific neutralizing antibody titers that reached or exceeded 0.5 IU/mL, with a geometric mean titer (GMT) of 0.96 IU/mL. This response was higher

than that of the inactivated vaccine (positive control), which had a GMT of 0.7 IU/mL (Fig. 3B, right). For the dual-immunogen RABV-G-LT mRNA vaccine (3 µg), only one mouse showed a neutralizing titer below 0.5 IU/mL at day 7 (Fig. 3B, right) (**Lines 373-380**, revised manuscript).” These results suggest that our mRNA vaccine candidates elicit a robust early immune response.

The related statements and figure legends have also been modified accordingly in the revised manuscript. (**Lines 67-72, lines 194-196, lines 373-380** revised manuscript).

3. *In the same way many of the comparisons were between the mRNA constructs when the comparison should be to current vaccines, if the co-authors wish to argue that their vaccine is a needed improvement.*

Response: We agreed with the reviewer’s comments. Indeed, the comparison between our RABV-G mRNA vaccine and current inactivated vaccine had been systemically implemented in our previous papers (A single vaccination of nucleoside-modified Rabies mRNA vaccine induces prolonged highly protective immune responses in mice. *Front Immunol.* 2023 Jan 17;13:1099991.), while this manuscript focuses on “How is the T-cell immunogen in addition to neutralization immunogen of RABV-G further improved the protective immunity against rabies”. We added sentences to further describe the primary goal for this study. We also provided the abstract from previous publication as following:

“ Background: Rabies is a lethal zoonotic disease that kills approximately 60,000 people each year. Although inactivated rabies vaccines are available, multiple-dose regimens are recommended for pre-exposure prophylaxis or post-exposure prophylaxis, which cuts down the cost- and time-effectiveness, especially in low- and middle income countries. Methods: We developed a nucleoside-modified Rabies

mRNA-lipid nanoparticle vaccine (RABV-G mRNA-LNP) encoding codon-optimized viral glycoprotein and assessed the immunogenicity and protective efficacy of this vaccine in mice comparing to a commercially available inactivated vaccine. Results: We first showed that, when evaluated in mice, a single vaccination of RABV-G mRNA with a moderate or high dose induces more potent humoral and T- cell immune responses than that elicited by three inoculations of the inactivated vaccine. Importantly, mice receiving a single immunization of RABV-G mRNA, even at low doses, showed full protection against the lethal rabies challenge. We further demonstrated that the humoral immune response induced by single RABV-G mRNA vaccination in mice could last for at least 25 weeks, while a two-dose strategy could extend the duration of the highly protective response to one year or even longer. In contrast, the three-dose regimen of inactivated vaccine failed to do so. Conclusion: Our study confirmed that it is worth developing a single-dose nucleoside-modified Rabies mRNA-LNP vaccine, which could confer much prolonged and more effective protection.”

4. *(Line 150) While the groups sizes varied the total number of mice used should be stated. Additional information regarding euthanasia criteria and euthanasia methods must be included for animal welfare. Were the animal experiments reviewed by an independent committee similar to IACUC?*

Response: We thank the reviewer for this critical comments. The total number of mice in each group has been re-sentenced to clear the confusion, so did for the according figure legends. We also revised manuscript to further address euthanasia criteria and euthanasia methods:

“ All mice experiments were conducted after approval by the Laboratory Animal Welfare and Ethics Committee of Shanghai Public Health Clinical Center (SPHCC) (approval number: 2021-A073-02). All infection experiments were performed in the ABSL2 laboratory following guidelines of Environmental Health and Safety. All work was performed with approved standard operating procedures and safety conditions for the RABV. The procedures used for anesthesia and euthanasia of study animals followed tenets of the ARRIVE reporting guidelines. During the RABV

challenge study, the mice with body weight drop more than 30%, severe paralysis, or inability to feed were euthanized using carbon dioxide inhalation”. (Lines 525-532, revised manuscript).

5. The discussion must be rewritten. It is currently too long and repetitive. The co-authors need to focus the discussion on their arguments and the supporting evidence.

Response: We thank the reviewer for this suggestion, we revised the the manuscript with a brief discussion as following.

“Rabies remains a devastating infectious disease globally, causing significant morbidity and mortality, especially in developing countries with limited vaccine access [1]. The currently available inactivated rabies vaccines, such as the 5-dose Essen regimen (1-1-1-1-1) and the 4-dose Zagreb regimen (2-1-1), effectively induce immunity. However, due to their poor immunogenicity [21, 30-33], the multiple-dose regimens imposes a substantial economic burden, limiting their use in low-resource settings [34, 35]. Therefore, developing a novel rabies vaccine that reduces costs while enhancing safety and efficacy is critical.

Recent advancements in mRNA vaccine technology offer a promising approach for developing a more effective rabies vaccine. This study introduces a novel non-replicating mRNA vaccine that encodes both the rabies virus glycoprotein (RABV-G) and a recombinant RABV-LT protein derived from the large protein (L). We first confirmed that RABV-LT is a strong immunogen, inducing T-cell responses and providing modest protection against lethal rabies virus. Then, by combining the RABV-LT T-cell immunogen with RABV-G mRNA, we developed the dual-immunogen vaccine RABV-G-LT. This vaccine showed superior control of lethal rabies virus infection in the brain and spinal cord compared to the inactivated vaccine or RABV-G mRNA alone, particularly in the early stages of vaccination. Additionally, RABV-G-LT facilitated significant recovery in body weight among infected animals, highlighting its therapeutic potential.

By targeting highly conserved epitopes within the L protein, we aimed to induce robust T-cell immune responses that enhance viral clearance. Utilizing bioinformatics,

we identified conserved epitopes from the L protein across various RABV strains and created a chimeric immunogen incorporating these elements. Our results confirmed that RABV-LT mRNA was immunogenic and elicited strong antigen-specific CD8+ T-cell responses, providing protection against lethal challenges. Our vaccine design emphasizes the importance of T-cell responses in combating viral infections, as these can prevent viral replication and eliminate infected cells. This concept is also supported by studies on other viral pathogens, including influenza [36] and SARS-CoV-2 [37, 38].

We further explored whether combining RABV-LT with RABV-G could provide improved protection against RABV, by incorporating an internal ribosome entry site (IRES) to promote the co-expression of RABV-G and RABV-LT from a single transcript. Additionally, a destabilizing domain sequence known as dihydrofolate reductase was located downstream of the RABV-LT sequence. This sequence has been shown to facilitate the rapid degradation of RABV-LT within the proteasome, enhancing epitope generation and antigen presentation to dendritic cells [39], thereby generating effective T-cell responses against infectious viruses [40, 41]. Our investigation revealed that the dual-immunogen RABV-G-LT provided stronger viral control and faster recovery from weight loss compared to inactivated vaccine or RABV-G mRNA, indicating enhanced protective efficacy. Importantly, the vaccine induced a Th1-biased immune response, critical for clearing virus-infected cells, reinforcing the notion that T-cell immunity is vital in defending against rabies.

Current vaccines are administered post-exposure to prevent disease onset, necessitating a prompt and robust immune response. Traditional inactivated vaccines have been enhanced with adjuvants, such as Toll-like receptor (TLR) agonists to accelerate immune responses [7, 12], or Mn J rabies vaccines to enhance both humoral and cellular immunity [42]. Our findings suggest that the bivalent RABV-G-LT mRNA vaccine could enable rapid production of virus-neutralizing antibodies (VNAs) and strong T-cell responses as early as seven days post-vaccination, positioning it as a potentially effective option for post-exposure prophylaxis.

Prolonged protection is essential for any vaccine. Our data demonstrated that the RABV-G-LT mRNA vaccine fully protected mice against lethal RABV challenges 20 weeks after a single vaccination, with RABV-LT-specific T-cell responses persisting beyond this period. Interestingly, while the molar dosage of RABV-G was the same, a 3 µg dose of the dual-immunogen RABV-G-LT vaccine resulted in diminished RABV-G-specific antibodies compared to a 1 µg dose of RABV-G alone. This suggests that RABV-LT responses may compete with RABV-G, reducing its specific antibodies. Understanding the regulatory mechanisms of RABV-G and LT-specific immunity following vaccination is crucial.

To evaluate the vaccine's performance in larger animals and humans, we tested the RABV-G-LT mRNA vaccine in non-human primates (NHPs). NHPs vaccinated twice at a 6-week interval with 3 or 10 µg doses of RABV-G or RABV-G-LT mRNA exhibited persistent antibody responses (titers above WHO-recommended thresholds) for at least 7 months. Notably, a third dose significantly enhanced VNA titers and reduced variability between the two vaccines, consistent with previous findings [43]. Assessing the RABV-G-LT mRNA vaccine's effectiveness in NHPs will help translate results more comprehensively to humans. Additionally, evaluating its efficacy in other animal models, such as hamster, ICR mice, beagles, and pigs, will provide further insights.

In conclusion, our study presents a promising new approach to rabies vaccination by combining the strengths of both humoral and cellular immunity through a novel mRNA vaccine platform, highlighting the urgent need for accessible and effective rabies prevention strategies globally.” (Lines 460-521, revised manuscript).

Minor comments:

6. (Line 41) *This statement is confusing. I believe the co-authors are referring to the onset of rabies symptoms once the virus reaches the CNS. In fact, the virus does not replicate rapidly after infection which is correctly stated in line 43.*

Response:

We apologize for this confusing statement, and revise the statement:

“It can paralyze the host's central nervous system (CNS) within a few days. Once

disease symptoms appear, the mortality rate approaches nearly 100%.” (Lines 57-58, revised manuscript).

7. (Line 50) Rabies PEP is highly effective and current vaccines are focused on inducing virus neutralizing antibodies because VNA is effective in preventing disease.

Response:

We thank the reviewer’s comment. We agree that rabies post-exposure prophylaxis (PEP) is highly effective and that current vaccines are designed to induce virus-neutralizing antibodies (VNA), which are vital for preventing the disease. However, our study emphasizes the importance of rapidly inducing both cellular immunity and neutralizing antibodies shortly after vaccination. Our findings indicate that while antibodies are indispensable, cellular responses are equally crucial for the vaccine to effectively promote virus clearance.

8. (Line 114) Were allergy and safety testing conducted in animals?

Response:

Thank you for the reviewer’s comment. We did not perform such testing in animals. However, to comprehensively assess the safety concerns associated with potential allergies induced by the LT antigen, we utilized five advanced predictive tools, including ALGPRED 2.0, AllerCatPro 2.0, AllerTOP v. 2.0, AllergenFP v. 1.0, and AlgPred 2.0-IgE. These tools allow us to conduct systematic predictive analyses to identify potential safety issues related to allergenicity. We believe that this approach provides a reasonable assessment of the safety of the LT antigen and lays a foundation for the future research. The result is shown below in the table, demonstrated the LT antigen was safety

Allergy Prediction Results of RABV-LT Antigen

Protein	AllerTOP V.2	AlgPred 2.0	AlgPred 2.0-IgE	AllergenFP V1.0	AllerCatPro 2.0
RABV-LT	NON	NON	NON	NON	No Evidence
Human insulin protein	ALLERGEN	ALLERGEN	ALLERGEN	ALLERGEN	Weak Evidence

9. (Line 127) *No mention of the FLAG tag.*

Response: We apologize for not describing FLAG tag, and revised the manuscript:

“Then, an internal ribosome entry site (IRES) from the encephalomyocarditis virus (EMCV) was introduced between the RABV-G and RABV-LT genes, creating a dual-immunogen mRNA, referred to as RABV-G-LT. The carboxy-terminal end of the RABV-G-LT gene includes an Escherichia coli dihydrofolate reductase (DHFR) domain, along with a FLAG tag for detecting RABV-LT expression.” (Lines 142-146, revised manuscript).

10. (Line 131) *add (see below) for protein expression and detection*

Response:

We thank the reviewer for pointing out the statement, and we revised as:

“The mRNAs were evaluated by agarose gel electrophoresis, and protein expression was determined in 293T cells (see below).” (Lines 149-150, revised manuscript).

11. (Line 174) *What expression system was used to make the G protein? Is it native, denatured, monomer, trimer?*

Response: The RABV-G protein used in this study was obtained commercially (ATAP10593, AtaGenix, China) and produced in HEK293T cells, allowing it to maintain its native trimeric structure.

12. (Line 236) *staining for RABV antigen would be more specific and informative than H&E*

Response: We have incorporated RNAscope in situ hybridization (ISH) to analyze rabies viral RNA expression in brain sections.

“RNAscope in situ hybridization (ISH) was performed to analyze rabies viral RNA expression in brain sections. The viral nucleoprotein (NP) RNA served as the target due to its high conservation across different rabies virus strains. For this assay, the NP-specific RNAscope probe (V-RABV-gp4 (220268) from ACDBio) was employed to detect viral RNA. The RNA ISH assay was conducted using the RNAscope Multiplex Fluorescent Reagent Kit V2 (Advanced Cell Diagnostics, 323100).” (Lines 255-259, revised manuscript).

The results are detailed below and added to the Fig. 3J:“ Correspondingly,we identified abundant dot signals of RABV-specific viral RNAs in the brains of empty-LNP mice, but no viral RNA in brain sections from vaccinated animals”. (Lines 420-422, revised manuscript).

The related statements and figure legends have also been modified accordingly in the revised manuscript. (Lines 255-259, lines 420-422, revised manuscript).

13. (Line 356) *Any animal euthanized with RABV in its brain it is difficult to say whether the animal is recovering or virus is just beginning to replicate and the animal will succumb. I think that including the current inactivated rabies vaccine for this comparison would be more informative since some animals did succumb in this group. Also this is a more important comparison than just comparing the two mRNA constructs.*

Response:

We apologize for not describing the comparison between the inactivated rabies vaccine and the mRNA constructs shown in Fig.3, which examined the differences of viral RNA level in protecting mice against RABV challenge, in the text. The description has now been added to the revised manuscript.

“Interestingly, our results demonstrate that both mRNA vaccine constructs outperformed the inactivated vaccine in promoting the reduction of viral RNA levels in the brain and spinal cord at different time-points. Specifically, the RABV-G-LT mRNA vaccine group exhibited an impressive approximately 20-fold reduction in viral RNA levels in the brain at 7 days post-inoculation (dpi), and a 7.2-fold reduction

at 12 dpi, when compared to the inactivated vaccine group (Fig. 3C, left). Similarly, the RABV-G mRNA vaccine also showed a viral RNA reduction in the brain or spinal cord; however, it was less effective, with reductions of only 3.2-fold and 2.1-fold at 7 and 12 dpi, respectively, in comparison to the RABV-G-LT mRNA vaccine (Fig. 3C, left). A similar trend was observed in the spinal cord, further reinforcing the superiority of the mRNA vaccine constructs over the inactivated vaccine (Fig. 3C, right).” (Lines 384-393, revised manuscript).

14. (Line 419-20) *Please cite evidence of poor immunogenicity of current rabies vaccines. The multi-dose regimen is more a product of the history of rabies PEP starting with Pasteur’s 21-dose regimen than any scientific evidence.*

Response:

We thank the reviewer for this instructive point. To address this point, we have revised and added the following statements to the Discussion section (Lines 461-465, revised manuscript):

“The currently available inactivated rabies vaccines, such as the 5-dose Essen regimen (1-1-1-1-1) and the 4-dose Zagreb regimen (2-1-1), effectively induce immunity. However, due to their poor immunogenicity [21, 30-33], the multiple-dose regimens imposes a substantial economic burden, limiting their use in low-resource settings [34, 35].” (Lines 461-465, revised manuscript).

15. (Line 480 and 499) *I would suggest the hamster rabies PEP model as a well-established model for further testing. Dogs and pigs should only be used if the vaccine is intended for vaccination of these species.*

Response:

We thank the reviewer for this instructive suggestion. We acknowledge that the hamster model is well-established for evaluating rabies vaccines and can provide valuable insights. And we will consider the use of the hamster rabies post-exposure prophylaxis (PEP) model for further testing in the future.

16. *Figure 1B the colors don’t match the legend so it is difficult to distinguish which line is which.*

Response:

The reviewer's comment is well taken. We have corrected the manuscript to ensure that the figure accurately reflects the details provided in the text. The revised Figure 1B is shown below.

The related statements and Figure 1 legends have also been modified accordingly.

17. Figure 1D add Immune analysis $n = 5$ / CVS-11 Challenge $n = 10$ (Line 640) correct $n=15$

Response: We apologize for this confusion. In our mice studies, we immunized 15 mice per group, 5 out of 15 mice were sacrificed first to assess T-cell responses, the left 10 mice per group were then used to determine the protective efficacy of RABV-LT mRNA vaccine against rabies viral challenge. Therefore, the discrepancy arises because Figure 1D reflects the total of 15 mice used in the mice study.

To ensure the figure precisely reflects the details provided in the text. We revised Figure 1D as below.

The description and figure legends have also been revised accordingly.

Point-by-point Response to Reviewers' Comments

Reviewer #1. Line 49-50: *This line was corrected, however the way it is stated right now implies that the current vaccines are unreliable, which is not the case shown by the number of people surviving thanks to PEP treatment. It would be an advantage to develop a vaccine which can develop sufficiently high antibody titers after single shot, and this would be an improvement, but I do not believe that you can state the current vaccines are unreliable.*

Response: We thank the reviewer for the comments. We agreed with the perspective and revised the statements as following:

"Rabies remains a devastating and fatal infectious disease globally. While current vaccines are effective, they require multiple administrations to achieve robust protective immunity, thus it is necessary to develop a more cost-effective vaccine."
(Lines 49-51, revised manuscript).

2. Line 57-58: *The rabies virus "infects" the central nervous system, it does not paralyze it. Paralysis is a potential result of the infection.*

Response: We apologize for this statement, and revise it:

"From the bite site, RABV travels along the peripheral nervous system to reach the central nervous system (CNS), and once clinical symptoms appear, the mortality rate approaches nearly 100%. "(Lines 58-60, revised manuscript)

3. Line 67-70: *Taken into account that it takes approximately 7-10 days for the immune system to elicit an immune response upon vaccination, do you really believe that a vaccine could be developed that makes the administration of immunoglobulines unnecessary? Passive immunoglobulines are administered at the side of exposure immediately or as soon as possible after exposure. With vaccines it would still take 7-10 days to have antibodies that can react...*

Response: We agreed with the reviewer's comments. After exposure, passive immunoglobulines should be administered at the site of exposure as soon as possible. And we revised the statements as following:

"Current vaccinations primarily focus on inducing effective neutralizing antibodies, but

cellular immunity is also crucial for controlling the spread of infection.” (Lines 65-67, revised manuscript).

4. Line 463: *Although I can understand that mRNA vaccines might elicit a higher immune response, I do not agree with stating that the current vaccines are poorly immunogenetic. They also elicit high and long lasting immune responses. I believe that the focus should be more on the fact that multiple-dose regimens results in therapy failure because of availability.*

Response: We agreed the reviewer’s comment and revised as following:

However, to achieve protective immunity, multiple-dose regimens are required, thus imposing a substantial economic burden and limiting their use in low-resource settings. (Lines 461-463, revised manuscript)

5. Line 500-502: *when being exposed to a potentially rabid animal, post-exposure, including HRIG should be administered as soon as possible. Waiting 7 days for an immune response via vaccination leaves the virus uncontrolled at the site of exposure. Do you really believe that this would be a safe solution?*

Response:

We appreciate the reviewer’s suggestions and revise the statements as following:

“Our findings suggest that the bivalent RABV-G-LT mRNA vaccine could enable rapid production of virus-neutralizing antibodies (VNAs) and strong T-cell responses as early as seven days post-vaccination, making it a potentially effective strategy for rabies prevention.” (Lines 497-500, revised manuscript).

Reviewer #2: *The co-authors have improved the manuscript and addressed all concerns and comment except I ask that they include the predictive tools used on line 133.*

Response: We thank the reviewer for the comments, and we incorporate and cite these tools in the revised manuscript as following.

“Then, allergenicity and safety tests were conducted using AllerTOP V.2、 AlgPred 2.0 、AlgPred 2.0-IgE、AllergenFP V1.0 and AllerCatPro 2.0.” (Lines 129-131, revised manuscript)